# The Nanoscopic Organization of Synapse Structures: A Common Basis for Cell Communication

**DOI:** 10.3390/membranes11040248

**Published:** 2021-03-30

**Authors:** Xiaojuan Yang, Wim Annaert

**Affiliations:** VIB Center for Brain and Disease Research and KU Leuven, Department of Neurosciences, Gasthuisberg, B-3000 Leuven, Belgium; xiaojuan.yang@kuleuven.be

**Keywords:** central synapses, neuromuscular junctions, immune synapses, nanodomains, super-resolution microscopy, STED, SIM, SMLM

## Abstract

Synapse structures, including neuronal and immunological synapses, can be seen as the plasma membrane contact sites between two individual cells where information is transmitted from one cell to the other. The distance between the two plasma membranes is only a few tens of nanometers, but these areas are densely populated with functionally different proteins, including adhesion proteins, receptors, and transporters. The narrow space between the two plasma membranes has been a barrier for resolving the synaptic architecture due to the diffraction limit in conventional microscopy (~250 nm). Various advanced super-resolution microscopy techniques, such as stimulated emission depletion (STED), structured illumination microscopy (SIM), and single-molecule localization microscopy (SMLM), bypass the diffraction limit and provide a sub-diffraction-limit resolving power, ranging from 10 to 100 nm. The studies using super-resolution microscopy have revealed unprecedented details of the nanoscopic organization and dynamics of synaptic molecules. In general, most synaptic proteins appear to be heterogeneously distributed and form nanodomains at the membranes. These nanodomains are dynamic functional units, playing important roles in mediating signal transmission through synapses. Herein, we discuss our current knowledge on the super-resolution nanoscopic architecture of synapses and their functional implications, with a particular focus on the neuronal synapses and immune synapses.

## 1. Introduction

The term “synapse” means “conjunction” and first appeared in 1897 in a textbook to describe the hypothesized connections between neurons in the nervous system [1], and which are nowadays known as central synapses. The actual existence of synapses in the central nervous system and their fine structural features were later characterized in detail with electron microscopy (EM) [2,3]. The concept of the neuronal synapse was further extended to include the communication from a neuron to another cell type, such as muscle cells [4]. These motor neuron-to-muscle cell synapses are also called neuromuscular junctions (NMJs). Furthermore, lymphocytes (including T and B cells and natural killer cells in the immune system) form cell-to-cell contacts with antigen-presenting cells (APCs) that resemble the neuronal synapses with respect to overall morphology, thus termed immunological synapses or immune synapses (IS) [5,6,7]. Bearing a similar name, “synapse”, these different types of synapses share similar characteristics in their structural organization. In general, they are cellular adhesions of stable plasma membrane segments of individual cells, mediating directed signal transmission [8,9]. Herein, generally, three compartments are distinguished: pre-synaptic compartment, synaptic cleft, and post-synaptic compartment.

The synaptic cleft is the space between the two plasma membranes (Figure 1). For central synapses, the cleft measures on average 20–30 nm along the perpendicular axis by EM [2,3]. The clefts of NMJs are roughly 30–50 nm [10], whereas the immune synapses are about 10–30 nm [11,12]. Though with variations, the synaptic clefts of different synapse types bear in common their similarly narrow spaces. On the other hand, the lateral dimensions of neuronal and immunological synapses differ largely (Figure 1). Central synapses are generally below 1 µm [13,14]. At NMJs, the motor nerve terminals form numerous varicosities (also called “boutons”, are generally 1–5 µm wide), and in total, can span 10–60 µm laterally [15,16]. The immunological synapses can also extend to several micrometers (5–10 µm) in diameter [17].

In addition to structural similarities, different types of synapses also share evolutionarily similar proteomes, such as cell adhesion molecules like cadherins and soluble N-ethylmaleimide sensitive-factor attachment protein receptors (SNAREs) that mediate vesicle fusion [21,22,23]. Given these structural and proteomic similarities, we consider that it can be insightful to discuss the nanoscale architecture of neuronal synapses and immunological synapses in the same context, particularly of super-resolution fluorescence microscopy (SRFM). Even though EM offers superior resolving power and has provided enormous advances in studying biological structures, it is limited by the substantial sample preparation procedure, the extreme imaging conditions, and the limited protein specificity with immuno-gold labeling. On the other hand, SRFM offers high throughput and compatibility for live-cell imaging, resulting in a wealth of new discoveries in the cellular ultrastructure as well as dynamics [24,25,26]. 

In conventional fluorescence microscopy, the resolution is limited by the diffraction and defined as 0.61 λ/NA according to the Rayleigh criterium, where λ is the wavelength and NA is the numeric aperture of the objective. This renders a resolution of approximately 250 nm, which often fails to resolve subcellular structures in detail. Several super-resolution techniques overcome this diffraction limit by different methodologies and achieve a much-enhanced resolution. Structured illumination microscopy (SIM) uses a series of spatially structured excitation light patterns to achieve a two-fold enhanced resolution [27]. Stimulated emission depletion (STED) microscopy reduces the focal spot by stimulated depletion at the rim, thus achieving a five-fold improvement of resolution [28,29]. Single-molecule localization microscopy (SMLM) comprises of several techniques, including stochastic optical reconstruction microscopy (STORM) [30], direct STORM (dSTORM) [31], photo-activated localization microscopy (PALM) [32,33], and points accumulation for imaging in nanoscale topography (PAINT) [34]. Though these SMLM techniques manipulate the fluorescent probes differently, they all enable the detection of single fluorophores and achieve resolution at a single molecule level of 10–40 nm [35]. 

In this review, we focus on the nanoscale structures of the central synapses in the nervous system, the NMJs between motor neurons and muscle cells, and the immune synapses. Previous studies on these synapses with confocal microscopy and EM, including freeze-fracture, have also provided enormous information on their (ultra) structures [36,37,38]. Here we discuss the recent discoveries in the ultrastructure of these synapses as analyzed in particular through the more recently developed SRFM technologies. By illustrating the similarities in the nanoscale organization of these different types of synapses, we argue that a common structural basis of synapses may exist for cell-to-cell communication and that researchers can find inspirations by comparing these different types of synapses.

## 2. The Neuronal Synapse

Neuronal synapses are the conjunctions between a neuron and another cell (either a neuron or another cell type, such as a muscle cell). Here, we particularly discuss the central synapses, which are formed between two neurons in the central nervous system (Figure 1D), and the NMJs, which are formed between motor neurons and muscle cells (Figure 1E). 

### 2.1. Central Synapses

#### 2.1.1. Laminar Organization at Post-Synaptic Density

Central synapses appear as small puncta in conventional fluorescence microscopy due to their small size. EM of immuno-gold labeled proteins has shown that specific proteins at the post-synaptic density (PSD) are arranged in different layers along the perpendicular axis of the synapse, with PSD-95 closer to the plasma membrane (~12 nm) and other proteins such as Shank and CaMKII closer to the cytoplasmic side (~24–26 nm) [39,40,41]. Proteins closer to the plasma membrane (such as PSD-95) have lower recovery rates/dynamics compared to those located more to the cytoplasmic side (such as Shank), as revealed by fluorescence recovery after photobleaching (FRAP) experiments [42]. With multi-color 3D STORM, Dani and colleagues have recapitulated this laminar organization at the PSD, with a much higher throughput of synapses and synaptic proteins [43]. This laminar organization and the dynamics of PSD proteins demonstrate the functional importance of the spatial distribution of synaptic proteins.

#### 2.1.2. Sub-Synaptic Domains (SSDs) in the Post-Synaptic Density 

It was previously believed that PSD scaffolding proteins were evenly distributed in the lateral planes of the PSDs, based on EM observations [39,40,44,45]. However, and thanks to SRFM techniques, it is now realized that many synaptic proteins are distributed heterogeneously and form nanodomains within the synapses, termed sub-synaptic domains (SSD) [14]. SSD can be defined as a sub-compartment of a synapse in which the density of a specific synaptic protein is higher than in the surrounding area and which is typically observed in super-resolution microscopy [14]. Therefore, synaptic proteins forming SSDs indicate, in essence, their heterogeneous spatial distribution within the synapse. It is important to keep in mind that large populations of proteins reside in non-SSD areas, and all of them go through dynamic changes. PALM imaging revealed that the major post-synaptic scaffold proteins at excitatory synapses (including PSD-95) and inhibitory synapses (including gephyrin) formed distinct SSDs [46,47,48]. The size of PSD-95 SSDs are ~80 nm in diameter [46]. This SSD organization of PSD-95 was also observed by STED microscopy of synapses *in vitro* [49,50] and *in vivo* [51,52,53]. The scaffold protein at inhibitory synapses, gephyrin, was also shown to form SSDs in studies with STORM, PALM, SIM, and STED microscopy [18,48,49,54]. Notably, these scaffold protein SSDs go through dynamic morphological changes both spontaneously and in response to activity changes. Time-lapse PALM imaging of PSD-95 revealed a continuous variation in the distribution of PSD-95 SSDs, and the number of PSD-95 SSDs per synapse was reduced by tetrodotoxin (TTX) treatment in primary hippocampal neurons [46]. Enlarged spine size resulting from chemically induced long-term potentiation (cLTP) was shown to associate with an increased number of PSD-95 SSDs *in vitro* by STED microscopy, and this increase occurred within hours in the time-lapse STED imaging [50]. Likewise, *in vivo* STED microscopy has shown the sub-structure and morphological changes in PSD-95 puncta within hours [53]. Therefore, these protein SSDs should be viewed as dynamically changing structures rather than rigid units. 

At the post-synaptic plasma membrane, neurotransmitter receptors have also been shown to be organized into SSDs by SRFM. PALM imaging has provided evidence that the excitatory α-amino-3-hydroxy-5-methyl-4-isoxazolepropionic acid receptors (AMPARs) are organized into SSDs of ~70 nm in diameter [55]. These AMPAR SSDs are as well dynamically changing in their composition, shape, and position and are regulated by PSD-95 expression levels [55]. 3D-SIM imaging of γ-aminobutyric acid type A receptors (GABA_A_Rs) at inhibitory synapses in the hippocampal and cortical neurons [54] and Dual-color dSTORM imaging of both the GABA_A_Rs and glycine receptors (GlyRs) in the spinal cord neurons [18] showed that these receptors are also organized into SSDs at inhibitory PSDs. Notably, the sub-domains of GlyRs were also observed in EM graphs [56].

Furthermore, synapses often accommodate different types of receptors at the same PSDs. In the neocortex and hippocampus, the ionotropic AMPARs and N-methyl-D-aspartate receptors (NMDARs) and the metabotropic glutamate receptor 5 (mGluR5) co-exist at excitatory synapses. A recent study using dSTORM showed that NMDARs formed a singular SSD mainly at the center of the PSD, whereas AMPARs segregated into several SSDs surrounding the NMDARs [57]. The differential distribution of these receptors detected by dSTORM is in line with the former observations using EM [58,59,60,61] and is consistent with the notion that NMDARs are less mobile than AMPARs at mature synapses [62,63]. On the other hand, mGluR5s are homogeneously distributed and broadly dispersed at the synaptic surface [57]. Similarly, mGluR4s at the parallel fiber active zones in the mouse cerebellum exist mostly as monomers or dimers [64]. As to the mixed inhibitory synapses, GlyRs and GABA_A_Rs are located at the same PSDs in spinal cord neurons [65,66]. Here they form distinct SSDs which are only partially overlapping, as shown by dual-color dSTORM [18]. These distinct SSD formations of GlyRs and GABA_A_Rs imply different underlying pathways for regulating glycinergic and GABAergic neurotransmission. The glycinergic and GABAergic co-transmission in spinal cord neurons is particular, given that GlyRs and GABA_A_Rs are activated by different neurotransmitters (glycine and GABA, respectively) released from the same pre-synaptic vesicles [67,68], whereas NMDARs, AMPARs, and mGluRs are activated by the same neurotransmitters (glutamate). However, the physiological consequences of this differential spatial organization and neurotransmitter activation of GlyRs and GABA_A_Rs at inhibitory synapses remain to be determined. Thus far, modeling data on excitatory synapses suggest that the distance between receptor SSDs to pre-synaptic vesicle release sites defines the number of activated receptors and shapes the neurotransmission response [46,57]. Given that GlyR- and GABA_A_R-mediated post-synaptic currents are kinetically different, their differential spatial organization at mixed inhibitory synapses may function alike at excitatory synapses to tune inhibitory neurotransmission. In addition, the heterogeneous content of glycine and GABA in pre-synaptic vesicles [68] may be another player to regulate inhibitory neurotransmission. 

#### 2.1.3. Sub-Synaptic Domains (SSDs) at the Pre-Synaptic Compartment 

The studies on the pre-synaptic compartment have mainly focused on the machinery responsible for neurotransmitter release from synaptic vesicles, including the cytomatrix active zone (CAZ) proteins, SNARE complexes, and voltage-gated calcium channels (VGCCs) [69,70]. SRFM data of pre-synaptic compartments, though relatively limited, have revealed the nanodomains/SSDs of these pre-synaptic proteins, as well as the dynamics of synaptic vesicles during the endo/exocytosis [71]. With single-particle tracking PALM (sptPALM), it has been shown *in vivo* that syntaxin1A (one of the glutamine- or Q-SNARE proteins located on the plasma membrane) is organized into nanoclusters/SSDs, of which the size and molecular density is regulated by activity and the general anesthetic propofol [72,73]. Similarly, VGCCs are also mobile yet dynamically confined into nanoclusters/SSDs, which are regulated by network activity in synaptic plasticity [74,75]. This SSD organization of the Ca_v_2.1 calcium channel was also observed in SDS-digested freeze-fracture replica labeling EM [76]. In addition, numerous CAZ proteins were shown to form SSDs by 3D STORM analysis, including Bassoon, Rab3-interacting molecule (RIM), and Munc13 [77]. Dual-color STORM also highlighted that the nanoclustering of CAZ proteins (Bassoon and RIM) was bidirectionally regulated by neuronal activity [78]. Herein, long-term blockade of activity by TTX induced reversible CAZ proteins un-clustering and local VGCC recruiting, suggesting a homeostatic regulation by the clustering status of CAZ proteins [78]. Another STED study of the pre-synaptic vesicle-associated protein synaptotagmin I revealed that it remained clustered at the plasma membrane after vesicle exocytosis, suggesting that they might be recycled together [79]. 

#### 2.1.4. Trans-Synaptic Nanocolumns

From another perspective, it is of great interest to know whether the synaptic vesicle release sites are aligned with the post-synaptic receptor SSDs, because their spatial relationship may shape the neurotransmission efficacy [80]. To resolve the myth, Tang and co-workers were first to identify that RIM SSDs predict the vesicle release site distribution and further demonstrated that post-synaptic AMPAR SSDs and scaffold protein (such as PSD-95 and Homer) SSDs are aligned with the pre-synaptic RIM SSDs [77]. Thus, they hypothesized the formation of trans-synaptic nanocolumns which guide the neurotransmitter release to occur near the receptors [77,81]. This trans-synaptic alignment was also observed *in vitro* and *in vivo* by STED microscopy [50]. Remarkably, the number of the nanocolumns scaled positively with the enlarged spines induced by cLTP, and these nanocolumns remained aligned though with enhanced mobility induced by cLTP [50]. Similarly, 3D SIM of inhibitory synapses in hippocampal and cortical neurons revealed that post-synaptic gephyrin SSDs and GABA_A_R SSDs are aligned with pre-synaptic vesicular GABA transporter (VGAT) and RIM SSDs, also forming nanocolumns at inhibitory synapses [54]. At mixed inhibitory synapses in spinal cord neurons, the distinct GlyR and GABA_A_R SSDs are as well integrated into nanocolumns consisting of gephyrin and RIM SSDs as revealed by dual-color STORM [18]. These trans-synaptic nanocolumns may therefore provide a novel genuine mechanism for regulating synaptic transmission and plasticity [81,82,83]. 

Given that synapses are dynamic rather than static structures [84], the next question is how are the pre-synaptic and post-synaptic SSDs aligned together and maintained through synaptic changes? EM data have shown the existence of discrete trans-cleft proteinaceous filaments [85,86,87], pointing to the synaptic adhesion molecules as a potential organizer for trans-synaptic nanocolumns. Neurexin and neuroligin complexes are potential candidates due to their important roles in synaptic functions [88,89]. Indeed, neuroligin-1 was shown to be tightly associated with AMPAR SSDs, and truncated neuroligin-1 shifted the alignment between RIM and AMPAR SSDs, impairing synaptic transmission [90]. However, combing several SRFM techniques (STED, STORM, and universal PAINT or uPAINT), Chamma and co-workers demonstrated that neuroligin-1 had a disperse distribution whereas leucine-rich repeat (LRR) transmembrane protein 2 (LRRTM2) was organized into SSDs at the post-synaptic membranes [91]. On the other hand, the pre-synaptic neurexin-1 displayed a dual-pattern distribution [92]. Given that both LRRTM2 and neuroligin-1 are binding partners for neurexin-1, the interaction between neurexin-1 and LRRTM2 has been proposed to potentially function as the nanocolumn organizer. Further evidence with STORM indicated that neurexin-1 was organized into nanoclusters at excitatory synapses that were dynamically regulated via ectodomain cleavage through a disintegrin and metalloproteinase domain-containing protein 10 (ADAM10) [93], supporting the notion of neurexin as a functional organizer for trans-synaptic nanocolumns. Alternatively, synapses may incorporate different adhesion molecules to account for the widely various synaptic changes. Ephrin receptors and their corresponding ephrin ligands are both membrane-bound proteins that have important functions in synapse formation, among many other cellular processes [94]. Given that Ephrin type-B receptor 2 (EphB2) was observed at synapses and formed nanodomains within the PSD area by STED microscopy [95], EphB2 and ephrin interaction may be the next candidate as nanocolumn organizers. Further investigation of EphB2 and ephrin distribution at synapses by dual-color SRFM, together with functional studies such as the effect of knock-out or overexpressing EphB2 on synaptic structures, may provide more insights into their potentiality as nanocolumn organizers. Identifying the trans-synaptic organizers will be key to understanding the dynamic changes of these synaptic ultrastructures. Herein, more recently developed proximity labeling techniques, including APEX2 and TurboID [96,97,98,99] followed by mass spectrometry analysis, could be instrumental in identifying such synapse-specific organizers. 

### 2.2. Neuromuscular Junctions (NMJs)

NMJs feature the numerous varicosities at the motor neuron terminus, or boutons, which accommodate several active zones at each bouton, and the folding of muscle cell plasma membranes which are called junctional folds [100,101]. The pre-synaptic active zones in EM graphs appear as T-bars in Drosophila NMJs or as small aggregates in vertebrate NMJs (Figure 1E). Given their relatively large size, NMJs (especially of *Drosophila*) have been the model synapses for understanding the pre-synaptic vesicle release mechanisms during neurotransmission. The *Drosophila* NMJs have been subjects of several SRFM based studies, providing protein identities in addition to the ultrastructure of NMJs with novel functions allocated to these proteins. Excellent recent reviews have been published elsewhere on the ultrastructure of NMJs [71,102,103], and here, we summarize the most important aspects. Briefly, most of these studies have focused only on the pre-synaptic compartment of NMJs, revealing the nanodomain distribution pattern of proteins responsible for synaptic vesicle release. In *Drosophila* NMJs, the coiled-coil domain protein Bruchpilot has been observed in donut-shaped structures centered at the active zone by STED microscopy [104]. Further dSTORM analysis revealed substructural units of the active zone containing an average of 137 ± 29 Bruchpilot proteins [105]. Each active zone could contain different numbers of these units, which were linked to the active zone physiological states and neurotransmitter release [105]. STED microscopy also revealed the distinct sub-active zone patterning of the two isoforms of Unc13A and Unc13B, with the former located at 70 nm away from the active zone center marked by Bruchpilot signal and the latter 120 nm away [106]. The same group further analyzed that Unc13A was positioned by Bruchpilot and RIM-binding proteins (RBPs) and that its precise positioning and local levels affect the release site number, position as well as functionality [106,107]. Based on these observations, a model of molecular nanostructure coupling for synaptic vesicle release has been proposed where scaffold proteins establish nanodomains that connect VGCCs to synaptic vesicles, thus influencing the probability of synaptic vesicle release [108]. This nanodomain patterning of synaptic proteins in NMJs is reminiscent of the SSDs in central synapses. During long-term synaptic potentiation of NMJs, it was shown by STED microscopy that the number of nanodomains or the quantity of proteins in each nanodomain was increased [109], supporting a quantal addition model for regulating synaptic strength and plasticity. This quantal addition model has also been proposed for the central synapses [81,82].

At vertebrate NMJs, dual-color STED microscopy revealed that the pre-synaptic Bassoon and Piccolo nanodomains are organized into Piccolo-Bassoon-Piccolo sandwich-like structures and that VGCC nanodomains colocalize with Bassoon nanodomains [110]. The same study also uncovered that Bassoon and VGCC protein levels were significantly decreased during aging, while Piccolo protein levels remained relatively stable. Therefore, selective degeneration of specific active zone proteins has been proposed as an aging process [110,111]. While most studies focused on the pre-synaptic nerve terminus of NMJs, York and Zheng examined the post-synaptic compartments of muscle cells adopting both SIM and STORM [19]. Their study revealed that the acetylcholine receptors (AChRs) were not uniformly distributed on the crest of junctional folds as previously thought. Instead, AChRs were concentrated at the edge of the crest, segregated from the integrin clusters that are more located towards the center of the crest [19]. It has been known for a long time that the pre-synaptic active zones of NMJs are aligned to the opening of the junctional folds [100]. This new finding with SRFM, therefore, suggests that the trans-synaptic alignment of AChRs to the active zones potentially enables effective synaptic transmission. Laminin in the basal lamina of NMJs may be the key player for the trans-synaptic alignment at NMJs, given that loss of laminin resulted in misalignment of active zones with junctional folds [112,113]. Taking together with the trans-synaptic nanocolumn in central synapses, the nano-alignment of post-synaptic receptors to pre-synaptic vesicle release sites may constitute a common mechanism underlying synaptic neurotransmission. 

## 3. Immune Synapses

Immune synapses are formed during T cell activation when T cell receptors (TCRs) engage with the antigen-presenting major histocompatibility complex (MHC) on the antigen-presenting cells (APCs). T cell activation is a dynamic process involving the recruitment of TCR microclusters into the center of the immune synapse, forming a mature immune synapse with a “bull’s eye” structure [114,115,116]. Specific molecules at mature immune synapses segregate into supramolecular activation clusters (SMACs). Laterally from inner to outer direction, immune synapses can be divided into three major domains, i.e., the central SMAC (cSMAC) enriched in TCRs, a peripheral SMAC (pSMAC) enriched in adhesion molecules such as lymphocyte function-associated antigen (LFA), and, finally, the distal SMAC (dSMAC) with F-actin forming dynamic protrusions [117,118]. This main architecture of immune synapses has already been drawn up by conventional microscopy thanks to their large size (5–10 µm). Interestingly, the segregation within the immune synapses, though on a much larger scale, is reminiscent of the segregation of AMPARs and NMDARs at small excitatory synapses in the central nervous system (see above).

The advent of SRFM offered a new opportunity to uncover the mechanisms underlying T cell activation. One of the hypotheses to be tested was that TCRs are organized into nanoclusters at the T cell resting state [119,120,121,122]. Evidence from PALM and STORM imaging pointed to the nanoclustering of TCRs and downstream signaling molecules at resting T cells [123,124,125]. However, other studies using different single-molecule-based fluorescent microscopy to analyze TCRs on live T cells suggested monomeric TCRs at resting T cells [126,127]. Discrepancies have also been observed on the adaptor linker for activation of T cells (LAT). One study using PALM and dSTORM showed that LATs on plasma membrane were neither phosphorylated nor recruited to TCR activation sites; instead, LATs were recruited through subsynaptic vesicles [128]. Another study using confocal imaging and tracking of LATs revealed that cell surface LATs were efficiently recruited within seconds of TCR engagement [129]. These discrepancies can at least be partially explained by the limitations inherent to SMLM [130] and the different sample preparation methods [131]. 

3D-SIM and STED microscopy have enabled the study of the dynamics and ultrastructure of the cytoskeletal actin at the immune synapse [20,132,133,134,135]. The actin skeleton is divided into four discrete actin networks, i.e., the lamellopodia-like branched actin in the dSMAC, the lamella-like actomyosin arcs in the pSMAC, the hypodense F-actin in the cSMAC, and the actin foci in the dSMAC and pSMAC [136,137]. These discrete actin networks create distinct functional regions at the immune synapse. To be noted, the actin cytoskeleton also plays important roles in the central synapses [138,139], which highlights the universal roles of actin networks. Furthermore, and combining SMLM and total internal reflection fluorescent (TIRF) microscopy, it has been shown that TCRs are mainly localized to the T cell microvilli and barely on the cell body [140]. This peculiar localization of TCRs at the microvilli may explain the discrepancies in the studies analyzing TCR nanoclusters. A modeling study further suggests that ligand discrimination by TCRs is likely based on these microvilli [141]. In addition, a recent study using STORM has identified TWIK-related acid-sensitive potassium channel 2 (TASK2) as a novel player at human immune synapses [142]. These new findings open new directions for investigating the mechanisms underlying T cell activation [143,144].

To be noted, SRFM has also been widely applied to other cell communication structures, namely epithelial cell junctions such as tight junctions, gap junctions, adherens junctions, desmosomes, and hemidesmosomes [145,146,147]. The superior resolution of SFRM techniques has enabled the detection of nanoclusters and subdomains of junction proteins, which could not be revealed with confocal microscopy. For example, a recent study using STED microscopy revealed nanoscopic segregation of polarity proteins in tight junctions, with PALS1-PATJ and aPKC-PAR6β present in an alternated pattern [148]. In the nervous system, electrical synapses (also called gap junctions) are also subject to SRFM. Two-color dSTORM of connexin 43 and aquaporin 4 in cryo-sectioned rat brain tissues has shown that aquaporin 4 is positioned at the edge of connexin 43 plaques [149]. Though not the focus of this review, these studies reinforce the promising power of SRFM in investigating the nano-architecture for cell communication. 

## 4. Perspectives

SRFM techniques have narrowed down the resolution gap between light microscopy and EM. They have facilitated breakthroughs in molecular cell biology, and life sciences aimed to decipher subcellular and molecular organization at nanoscale resolution. The promising capacity of SRFM manifests in data throughput, target specificity, superior resolution, and especially live-cell imaging compatibility, which enables the visualization of dynamic cellular processes. And all that has happened in only a decade of years. Emerging correlative approaches of SRFM with (cryo-)EM further combine their respective advantages, up to enabling the 3D analysis of nanoclusters/structures *in situ* [150,151,152]. In addition, other combined SRFM modalities, such as lattice light-sheet microscopy combined with SMLM [153] and various 3D super-resolution microscopy techniques [154,155], will also enhance the study of native nanoscopic structures. Due to their complexity in set-ups and usage, these advanced microscopy modalities currently mainly exist in some expert groups and institutional imaging cores. With more and more accessibility promoted by commercialization, these techniques are envisioned to be widely used and yielding novel discoveries. Another future perspective is to identify the molecular components within the nanoclusters/nanodomains. STORM has been used to identify the neurexin nanoclusters containing different neurexin isoforms in neurons [93]. SMLM, together with high spatial resolution proximity labeling assay like Split-TurboID [99], may provide concrete evidence for nanocluster content screening. 

The studies of synapse structures have benefited a lot from SRFM techniques and will continue to advance with the improving microscopy modalities. On the other hand, given the structural and molecular similarities of the various synapse structures, we argue that studies of one synapse type can shed light and inspire the studies of another synapse type. A necessary comparison between the discoveries in different types of synapses can help us to better understand the basis of cell communication, irrespective of the cell types. To extend this aspect, inter-organellar membrane contact sites (MCSs) are dynamic nanoscale structures where intracellular organellar communication takes place. SRFM techniques have also been developed for imaging these MCSs, such as endoplasmic reticulum (ER)-plasma membrane MCSs [156,157] and ER-mitochondria MCSs [158,159]. MCSs share the structural pattern of two membranes and a narrow cleft. Therefore, the methodologies in SRFM of synapse structures can be extended to the study of MCSs, and which will lead as well to a better understanding of the dynamics of inter-organellar communication.

## Figures and Tables

**Figure 1 membranes-11-00248-f001:**
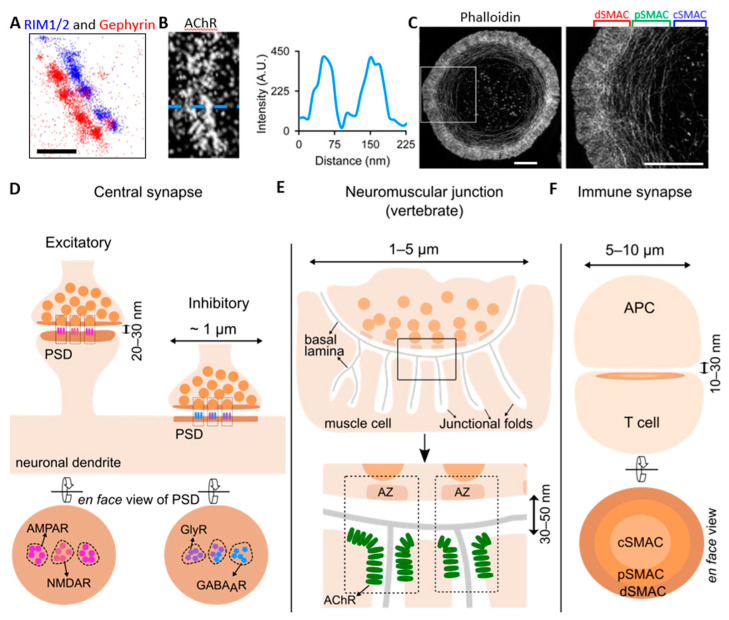
Synaptic structures under super-resolution microscopy. (**A**) Dual-color direct stochastic optical reconstruction microscopy (dSTORM) of pre-synaptic Rab3-interacting molecule 1/2 (RIM1/2) (in blue) and post-synaptic gephyrin (in red) at inhibitory synapses in cultured spinal cord neurons, showing the alignment of their nanodomains (modified from [18]). Scale bar: 200 nm. (**B**) dSTORM of acetylcholine receptor (AChR) strip at a neuromuscular junction (NMJ) (left) and line-scan profile (right) showing the slit in the AChR strip [19]. (**C**) 3D-structured illumination microscopy (SIM) imaging of an activated Jurkat T cell stained with phalloidin, showing the discrete actin networks [20]. Zoom-in view of the boxed region on the left is shown on the right. Scale bar: 5 µm. (**D**) Excitatory and inhibitory synapses in the central nervous system, with a size generally below 1 µm and synaptic cleft of 20–30 nm. Post-synaptic receptors are organized into sub-synaptic domains (SSDs) and aligned with pre-synaptic vesicle release sites, forming trans-synaptic nanocolumns (indicated by boxes with dashed lines in the upper panel). The lower panel shows the en face view of the excitatory and inhibitory post-synaptic density (PSD). The left shows the N-methyl-D-aspartate receptor (NMDAR) SSD in the center and several α-amino-3-hydroxy-5-methyl-4-isoxazolepropionic acid receptor (AMPAR) SSDs surrounding it at the excitatory PSD. The right shows the glycine receptor (GlyR) and γ-aminobutyric acid type A receptor (GABA_A_R) SSDs and their partial overlapping at the inhibitory PSD. (**E**) The vertebrate neuromuscular junction (NMJ), which has a diameter of 1–5 µm, and the synaptic cleft is 30–50 nm (lower panel). The muscle cell plasma membrane forms many junctional folds, and the synaptic cleft is resided by the basal lamina. The lower panel is the zoomed-in view of the boxed region in the upper panel. The dashed boxes indicate the trans-synaptic nanocolumns consisting of the pre-synaptic active zones (AZ) and the post-synaptic AChR clusters at the junction crest shoulder. (**F**) The immune synapse (IS) formed between T cell and antigen-presenting cell (APC), with a size of 5–10 µm and a cleft of 10–30 nm. The lower panel is the en face view of the post-synaptic compartment of the IS, depicting the central supramolecular activation cluster (cSMAC) enriched in T cell receptors (TCRs, light orange), the peripheral SMAC (pSMAC) enriched in linker for activation of T cells (LATs, orange), and the distal SMAC (dSMAC) enriched in F-actin (brown).

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
