# Peer review of "The Nanoscopic Organization of Synapse Structures: A Common Basis for Cell Communication"

_membranes, 2021, doi:10.3390/membranes11040248_

Round 1

Reviewer 1 Report

This review proposes to detailed organization of the synapse at the nanoscale level. The review is easy to read, but suffers a bit from an unbalanced organization: post-synaptic compartment of the central synapse is deeply described, but the description concerning presynaptic terminals and immune synapse or NMJ are still a bit short. Choosing to compare those synapses was to me an interesting initiative that could be further sustained with deeper data.

The notion of sub synaptic domains (SSD) is intensively described concerning post synaptic receptors and anchoring proteins of central synapse, as already reviewed previously by one of the author, but without defining the first SSD abbreviation in figure legend. Although this concept has been used by several teams concerning post-synaptic receptors and some presynaptic active zone protein, authors seems here to extend this concept to all synaptic proteins in a general way, which is to me a bit hazardous. Maybe rephrasing a bit the general sentences would help to refine the conclusion. Indeed one could expect for example, that presynaptic proteins associated with synaptic vesicles are not necessarily all clustered in SSD, but are rather following the positioning of synaptic vesicles which are clearly distributed among the whole bouton volume as seen by EM 3D data. It is interesting to note that SRFM data allowed identification of nanodomains for some proteins, but to be fair, proteins which have a non clustered distribution should also be mentioned.

From NMDAR and AMPAR organization (which display respectively a central organization within PSD for the former, and a lateral one for the later), the authors raise the conclusion that NMDAR are stably incorporated at PSD whereas AMPAR undergo dynamic exchanges with extrasynaptic receptors. This is an over simplistic shortcut since there is an extensive literature on NMDAR extrasynaptic localization, recycling and function outside synapse.

Figure concerning the organization of immune synapse is a bit less detailed than the 2 other components. Maybe adding schema on the TCR receptor and LAT distribution would decrease this unbalanced structure. Adding non SRFM data on the structure of the synapse (like concept coming from confocal or EM data) would also enriched the review.

Author Response

Reviewer #1:

This review proposes to detailed organization of the synapse at the nanoscale level. The review is easy to read, but suffers a bit from an unbalanced organization: post-synaptic compartment of the central synapse is deeply described, but the description concerning presynaptic terminals and immune synapse or NMJ are still a bit short. Choosing to compare those synapses was to me an interesting initiative that could be further sustained with deeper data.

We appreciate that the reviewer is interested in this comparison of the synapses described in this review paper. We agree with the somehow unbalanced organization, but this is merely reflecting a currently similar imbalance in the use of SFRM with respect to the different types of synapses. The standing point of this review is our own experience in super-resolution imaging of neuronal synapses, and by which we have noticed the comparable features of the different synapse types in super-resolution images. Also, most pre-synapse studies are done with NMJs. Secondly, comprehensive recent reviews have been published for the immune synapse (e.g. ref. 130, 131) and NMJ/pre-synapse (e.g. ref. 71, 102, 103) in the respective research fields. Within the page limit of the review, we felt that a focus on the use of SFRM in analyzing these structures fills up a gap in the literature. However, where possible we have included some more examples, such as for instance on the STED imaging of the presynaptic vesicle protein, synaptotagmin (lines 213-215), to have a more inclusive review on the use of SFRM in synapse structures. Newly added references are updated in the reference list. We hope that through our comparative approach experts from very different fields of the life sciences will be attracted to our review.

The notion of sub synaptic domains (SSD) is intensively described concerning post synaptic receptors and anchoring proteins of central synapse, as already reviewed previously by one of the author, but without defining the first SSD abbreviation in figure legend. Although this concept has been used by several teams concerning post-synaptic receptors and some presynaptic active zone protein, authors seems here to extend this concept to all synaptic proteins in a general way, which is to me a bit hazardous. Maybe rephrasing a bit the general sentences would help to refine the conclusion. Indeed one could expect for example, that presynaptic proteins associated with synaptic vesicles are not necessarily all clustered in SSD, but are rather following the positioning of synaptic vesicles which are clearly distributed among the whole bouton volume as seen by EM 3D data. It is interesting to note that SRFM data allowed identification of nanodomains for some proteins, but to be fair, proteins which have a non clustered distribution should also be mentioned.

We thank the reviewer for pointing out the undefined abbreviation SSD. We have added the full name with the first SSD (line 60 in the manuscript).

We thank the reviewer for the insightful comments on the clustered and non-clustered proteins. By clustered, the reviewer means that specific proteins form SSDs, as in contrast to non-clustered proteins which have diffusive or homogeneous distribution. This is in fact not what is meant with SSD. SSD are rather defined as a sub-compartment of a synapse in which the density of a specific synaptic protein is higher than in the surrounding area, and which is typically observed in super-resolution imaging (ref. 14). Therefore, synaptic proteins forming SSDs indicate in essence their heterogeneous spatial distribution within the synapse. It is indeed important to keep in mind that these proteins also reside in areas which are not detected as SSDs, such as areas surrounding the SSDs and peri- and extra-synaptic areas. To make this concept more clear, we have refined the introduction of SSDs (lines 134-139). We have also re-phrased some sentences (in lines 17, 132, 199) to make this point more clear.

Most synaptic proteins studied so far using super-resolution microscopy show heterogeneous distribution, hence forming SSDs, only with exceptions for metabotropic glutamate receptor (mGluRs), i.e. mGluR5 at excitatory synapses in hippocampus and mGluR4 at the parallel fiber active zones in the cerebellum. They are mentioned in the manuscript (lines 173-176).

From NMDAR and AMPAR organization (which display respectively a central organization within PSD for the former, and a lateral one for the later), the authors raise the conclusion that NMDAR are stably incorporated at PSD whereas AMPAR undergo dynamic exchanges with extrasynaptic receptors. This is an over simplistic shortcut since there is an extensive literature on NMDAR extrasynaptic localization, recycling and function outside synapse.

We thank the reviewer for pointing this out. The reviewer is correct about the dynamic exchanges of both AMPARs and NMDARs, not to mention that the dynamics is also observed for proteins residing in the SSDs. We apologize for having phrased it too simplified. The single particle tracking (SPT) experiments have revealed somehow more mobility of AMPARs than NMDARs at mature synapses (ref. 62, 63). The observation of AMPAR SSDs surrounding NMDAR SSD was therefore proposed in this study as potential explanation (ref. 57). We have re-phrased the related sentences to make this point more clear (lines 171-173).

Figure concerning the organization of immune synapse is a bit less detailed than the 2 other components. Maybe adding schema on the TCR receptor and LAT distribution would decrease this unbalanced structure. Adding non SRFM data on the structure of the synapse (like concept coming from confocal or EM data) would also enriched the review.

In the graph of immune synapses (Figure 1F, lower panel), the color codes are meant to represent different proteins. We realized that this was not mentioned in the legend, and we have completed the description (lines 70-72). Hopefully, this would add more information to the graph representation.

We agree with the reviewer that non-SRFM data have provided enormous information on the structure and even dynamics of these synapses. In this review, we aim to update the field with new discoveries by SRFM. We have made this point clearer in the text (lines 102-105). And non-SRFM data have been cited when necessary in this manuscript, including data from confocal microscopy, EM and electrophysiology. In addition, we have added more EM data (lines 102-104, 165, 206-207) to further connect our review with previous literature.

Reviewer 2 Report

In this review manuscript, Yang and Annaert compare neuronal (central synapses and NMJ) and immunological (APC/TC) synapses. While these two structures are seemingly different at first glance, they share many common features: tight protein complexes mediate alignment of plasma membrane contacts and swift activation of signaling cascades occurs at these specialized sites. The authors describe how the recent advances in cutting-edge fluorescence microscopy (STED, PALM/STORM) and mass spectrometry techniques (APEX, TurboID etc.) can help decipher the identity of the proteins at these cell-cell contacts and their dynamics during neurotransmission or immune cell activation. Together, this piece is well written, timely, and fits perfectly well for publication in Membranes.

Author Response

We gratefully appreciate the reviewer’s positive and approving comments.

Reviewer 3 Report

Review: The nanoscopic organization of synapse structures: a common 2 basis for cell communication.

This review focuses on the use of super-resolution microscopy techniques (STED, SIM and SMLM) to investigate super-resolution nanoscopic architecture of synapses and their functional implications, with a particular focus on the neuronal synapses and immune synapses.

This is well constructed review of the literature in synaptic structure revealed by microscopy techniques. The addition of reviewing the T cell receptors is interesting to demonstrate what has been revealed with these advanced microscopy techniques.

There does not appear to be any needed additions or changes required in this review as this is a very complete review and documented well with citations.

Only as a suggestion for the authors. They may wish address what the technique of freeze fracture has provided for high resolution of synaptic structure and proposed relation of structures to function.

Franzini-Armstrong, C. 1976 Freeze-fracture of excitatory and in-hibitory synapses in crayfish neuromuscular junctions, J. Microsc.Biol. Cell., 25: 217–222

Heuser, J.E. and Reese, T.S. 1973 Evidence for recycling of synapticvesicle membrane during transmitter release at the frog neuromuscu-lar junction, J. Cell Biol., 57: 315–344.

Cooper RL, Winslow JL, Govind CK, Atwood HL. Synaptic structural complexity as a factor enhancing probability of calcium-mediated transmitter release. J Neurophysiol. 1996 Jun;75(6):2451-66. doi: 10.1152/jn.1996.75.6.2451. PMID: 8793756.

Govind, C.K. and Meiss, D.E. 1979 Quantitative comparison of low-and high-output neuromuscular synapses from a motoneuron of the lobster Homarus americanus, Cell Tissue Res., 198: 455–463.

Author Response

We gratefully appreciate the reviewer’s positive and approving comments. We agree that the EM, including freeze fracture EM, has undoubtedly provided invaluable information on the synaptic ultrastructure. We have added more data using EM and freeze fracture EM to make the connection to SFRM (lines 102-104, 165, 206-207).